# Advances in Experimental Research on the Influence of Friction Powders on Acoustic Liners (Helmholtz Resonators)

Constantin Sandu, Andrei-George Totu *, Andrei-Tudor Trifu and Marius Deaconu

Romanian Research and Development Institute for Gas Turbines-COMOTI, 220D Iuliu Maniu Ave., 061126 Bucharest, Romania; constantin.sandu@comoti.ro (C.S.); andrei.trifu@comoti.ro (A.-T.T.); marius.deaconu@comoti.ro (M.D.)
* Correspondence: andrei.totu@comoti.ro

**Abstract:** This paper presents the technological advancement of using friction powders to increase the absorption of acoustic liners used in the reduction of tonal noise generated by aero-engines or for other applications related to Helmholtz resonators used in noise absorption of low frequencies. The experimental research was conducted during the European project ARTEM (2017–2022), and after. This concept was inspired by the discovery made by several historians of narrow neck bottles filled with ash in the old Christian churches. These artifacts were made with the purpose of absorbing low frequency noises. Based on this creative idea, the present authors proposed a new method of noise absorption capabilities of acoustic liners filled with various types and quantities of natural and artificial powders. Considering the positive results the ARTEM project offered, COMOTI continued testing this concept by using even finer cork powders manufactured with a new technology. Measurements in Kundt tubes showed that noise absorption increased significantly in broadband for low frequencies (over 0.9 at high frequencies and 0.6 at low frequencies, 500 Hz). Some of the researched powders can be used in the field of bladed machines to reduce the aerodynamic noise of an aircraft or in the automotive industry where the reduction of low frequency noises is necessary.

**Keywords:** acoustic liner; cork powder; Helmholtz resonator; friction powder

## 1. Introduction

It is known that high-performance aircraft engines generate fan noise which propagates through the intake [1]. In this case, the simplest noise reduction method is to use acoustic liners. These liners are comprised of numerous back-plated honeycomb structures, covered on the airstream side with a perforated sheet, that use the Helmholtz resonator principle to absorb certain frequencies (for example the rotation frequency of a fan's rotor). The acoustic liners are placed in the air intake of the engine and its bypass ducts [2].

However, the fan noise is a broadband one. For this reason, in the last decades the objective of the acoustic liners shifted to a multivalent one i.e., to be used not only for the absorption of one or two resonance frequencies but for the absorption of more frequencies. Research has shown that such a technical approach is possible. For example, EPFL (École Polytechnique Fédérale de Lausanne, Switzerland) researched the possibility of using plasma actuators to absorb the broadband low frequency noise. This solution achieved an absorption of broadband noises as a result of a partial ionization of a thin air layer with an atmospheric corona discharge and its control with an alternating electrical field [3–5]. ISVR (Institute of Sound and Vibration Research, Southampton) uses slanted perforated plates introduced in the resonance cavities of acoustic liners to absorb an even broader spectrum of noise frequencies [6].

In the European project ARTEM (Aircraft noise Reduction Technologies and related Environmental iMpact, Topic MG-1-2-2017, Call Identifier H2020-MG-2016-2017), in the WP1-Reduction Technologies for Sound Radiation, the technology of introducing friction powders into the resonance cavity of acoustic liners was researched [7]. The main idea was

to introduce fine friction powders into the acoustic liner cells (resonant cavities) to broaden the noise frequency absorption well. This idea was very efficient because it showed the existence of a new noise absorption phenomenon generated by the friction between the powder particles present in the resonance cavities of the acoustic liner. The absorption level was not influenced, but the powders widened the noise attenuation range. The phenomenon, as a result of friction, has specific characteristics different from the porous or fibrous material noise absorption mechanism [8,9]. The light weight and easy pourability of powders in the resonance cavities open a new way to reduce unwanted noise in the aviation industry and other fields where broadband noise absorption is necessary. The use of powders in acoustic liners is a relatively new idea, not widely addressed by the scientific community, with only one other significant similar paper identified by the authors [10]. Other solutions such as the use of bulk absorbing materials [11] have been identified in the literature, ranging from porous materials or foams [12] placed in the flow channel to acoustic liners filled with fibrous materials (e.g., metal fiber [13])

Surprisingly, the Helmholtz resonators (Figure 1a) were known and used many centuries ago.

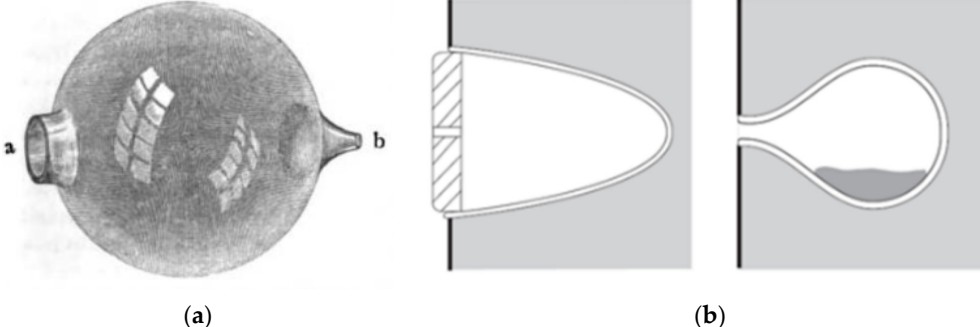

(a)        (b)

**Figure 1.** The Helmholtz resonator (**a**) Spherical shaped [14]; (**b**) Bottles made of glass filled with wood ash implanted in church walls for low frequency noise absorption (reduction of reverberation) [15].

These resonators were used at large scale in many old churches and Christian cathedrals. For example, it was found that for the reduction of reverberation, glass bottles filled with wood ash were placed in the walls of several churches in Denmark and Sweden (15–16-th century) (Figure 1b) [15]. These facts led to the idea that a special effect should happen when noise interacts with powder introduced into the acoustic liners, leading to a more efficient noise absorption in the fan duct of aero-engines. There are many other examples where this technology was used centuries ago, even from the Byzantium period [16]. In the modern era, it is known that noise has the power to act on powders. In his initial experiments, Kundt radiated a sound in a horizontal glass tube and he observed that the sound influenced the cork powder obtaining standing waves patterns. When Kundt adjusted the tube length for resonance, the powder was accumulated in the nodes of the standing wave.

The action of sound on fine particles is used at present for sound wavelength measurements in dusty environments [17]. All these facts show that sound strongly interacts with powders. From the point of view of measuring technologies, it is known that sound attenuation is currently used for dust concentration measurements in some present technologies [6]. This is the background which oriented the research during the project, and after. This phenomenon can be explained mainly by the friction between the powder granules which consumes the power of noise in a broad range of frequencies. The friction powders technology requires low manufacturing costs and can be used in multiple applications where absorption of low frequencies simultaneous with a certain resonance frequency is necessary. The theoretical aspects of this technology should be researched in the future.

## 2. Friction Powder Research

The present authors tested a total of 17 various materials in Kundt impedance tubes. Some materials were processed into powder/granules using an impact grinding machine and others were used as they were found on the market. These materials were as follows: micanite, polyurethane, spheres of expanded polystyrene, cork, balsa wood, colophony resin, talcum, Fomalux (plastic material), cinder, bird fluff.

Images of some of these powders/granules are presented in Figure 2. The apparent density of tested powders was between 0.0032 g/cm$^3$ and 0.7072 g/cm$^3$ depending on the nature of the material. The dimensions of particles generated by the impact grinding technology were irregular, distributed between 100 μm and 1000 μm.

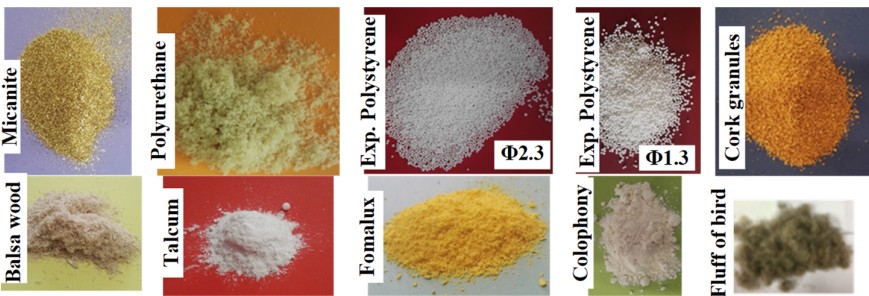

**Figure 2.** Various types of powder selected for tests [7].

The test sample dimensions (honeycomb and covers), used for Kundt tube absorption measurements were:

- Honeycomb height: 35 mm;
- Honeycomb cell size: 10 mm;
- Honeycomb wall thickness: 0.05 mm (aluminum);
- Honeycomb diameter (large samples): ΦL 100 mm;
- Honeycomb diameter (small samples): ΦS = 28 mm;
- Cover thickness: 0.7 mm;
- Hole diameter: Φ 1.8 mm;
- Porosity: 28.5%.

The cavities of the small (Φ28 mm) and large (Φ100 mm) test samples were manually filled with powders at 33%, 66%, and 100% of the honeycomb height. One small and one large sample were kept empty to act as reference test samples. A total of 84 samples filled with powders at the indicated percentages and two empty samples were prepared for measurements in the Kundt impedance tubes. The samples with the diameter of Φ28 mm were tested to see how they would behave in terms of acoustic noise reduction for frequencies in the range of 500–6300 Hz and the ones with the diameter of Φ100 mm were tested to measure their impact for the frequencies in the range of 100 to 1600 Hz. Finally, based on the experimental results, six powders were selected to have the potential of reducing noise generated by the onboard engines of aircrafts. The selection criteria were, maximum absorption of noise in broadband, minimum density of powder, low manufacturing costs and easy adaptation of the existent manufacturing technology for the acoustic liners to these kinds of powders.

The powders which presented the highest absorption coefficient in broadband were balsa wood powder, cork powder, expanded polystyrene balls (1.3 and 2.3 mm in diameter), Fomalux (plastic material) powder and grinded bird fluff. Looking at the obtained data [7], it is clear that noise absorption is high in broadband and depends on the nature of powders and the percentages of honeycomb filling with powder.

All six selected materials were tested for noise absorption in an air stream generated by a wind tunnel ('grazing flow') at CNRS (Laboratoire d'Acoustique de l'Université du Mans, Le Mans, France, LAUM-UMR CNRS) [18,19]. The tests were conducted using prismatic samples (200 mm × 49 mm × 34.1 mm) which were filled with powder at 33%,

66%, and 100% of honeycomb height in grazing flow conditions (M = 0 and M = 0.13). All the tested samples at the CNRS facility (Figure 3, two sources method) showed that cork powder (filling 66% of liner's height) performed the best. This was the starting point for the next test campaign, conducted at NLR, using various combinations of plates and honeycombs at different grazing flow speeds (Mach 0.2 and 0.4) in order to better understand the phenomenon.

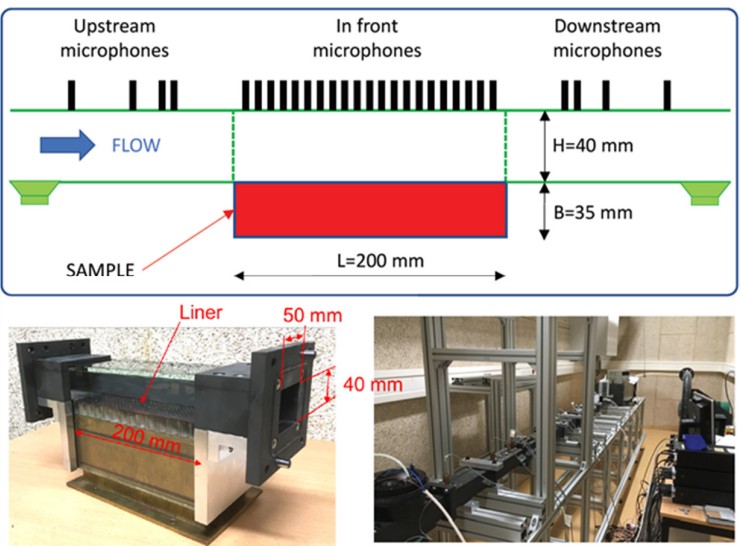

**Figure 3.** The CNRS wind tunnel for tests in 'grazing flow' [7].

For the NLR tests (Figure 4—experimental setup), four acoustic liner samples (obtained by changing the resonant cavity height) were prepared. The overall dimensions of the liner are 500 mm × 170 mm with varying heights of the resonance chambers (22.3 mm–42.4 mm).

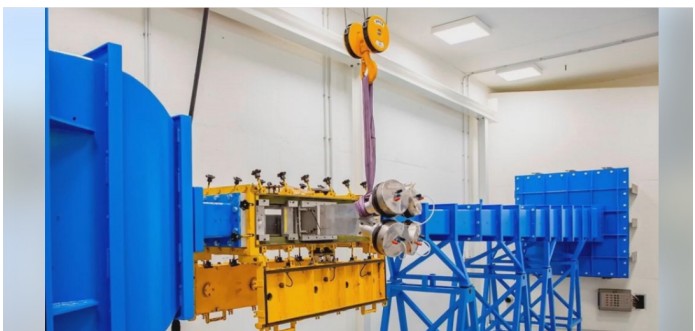

**Figure 4.** Engine liner testing wind tunnel of NLR FDF ("Flow Duct Facility") [20].

Two interchangeable perforated covers a and b were manufactured for the aforementioned acoustic liners:

(a) The first cover: thickness t = 1.5 mm, hole diameter d = 1.5 mm, distance between holes = 3 mm, porosity 23%; length × width = 500 mm × 170 mm
(b) The second cover: thickness t = 1 mm, hole diameter d = 1.5 mm, distance between holes = 2.5 mm, porosity 33%; length × width = 500 mm × 170 mm

The test bench used was designed to characterize the acoustic capabilities of reactive structures by measuring insertion loss. Insertion loss is defined as the difference between the pressure levels measured at a reference point before and after acoustic treatment. Thus, in the presented procedure, the IL results were obtained by the difference between the average sound pressure level inside the room without acoustic treatment (inside the grazing flow wind tunnel) and the average sound pressure level with acoustic treatment applied. This parameter implies an easier measurement procedure, as it describes very well the

acoustic behavior of a structure. As in [21], for each measurement and frequency band, the squared amplitude of the Fourier transformed sound pressure p2, is calculated from the measured sound pressure level SPL, with $p_{ref}$ = 20 µPa. Several measurements (using rotating microphones) are made in the sending and receiving room in order to correct the power level according to background noise. An example is plotted in Figure 5. For both lined and hard wall configurations, signal to noise ratio was, on the 0–5000 Hz range, above the imposed limits (>4 dB as acceptable and >7 dB as good—yellow and green dashed lines in Figure 5a).

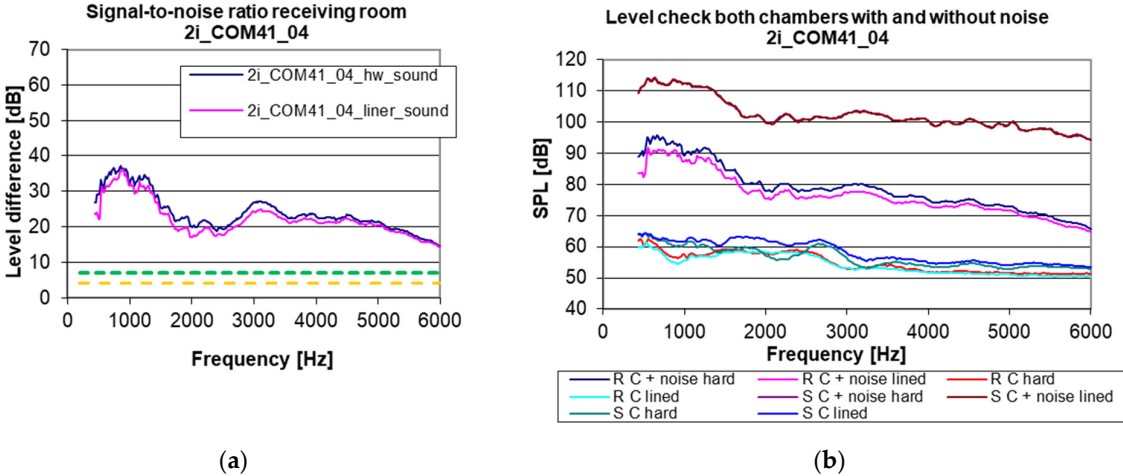

(**a**) (**b**)

**Figure 5.** Spectra of source/receiving room signals (Liner 4, cover A, Mach 0.4). (**a**) Signal-to-noise ratio receiving room 2i_COM41_04; (**b**) Level check both chambers with and without noise 2i_COM41_04.

Introducing powder into the actual liners (Figure 6) can lead to increases in attenuation at certain frequencies; however, a problem that can occur is a decrease in the reactance of the liners due to changes in the air volume properties inside the cavities. Therefore, this paper proposes fundamental research that should be further investigated for acoustic structures with lower porosities (<23%). The different combinations resulting from the resonant cavities and the two covers (for example Figure 7a), as well as the presence or absence of an acoustically permeable mesh that prevents the exit of the fine powder from the resonant cavity (Figure 7b,c), were experimented both in stationary mode (no flow) and in grazing flow conditions.

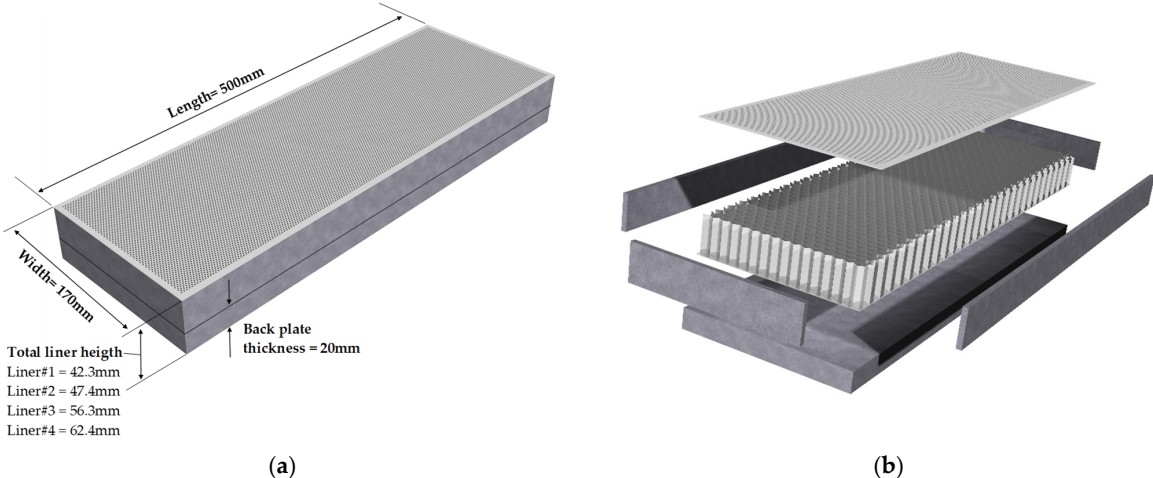

(**a**) (**b**)

**Figure 6.** The dimensions of the four acoustic liners used for testing wind tunnel of NLR ((**a**) Overall dimensions, (**b**) Exploded view).

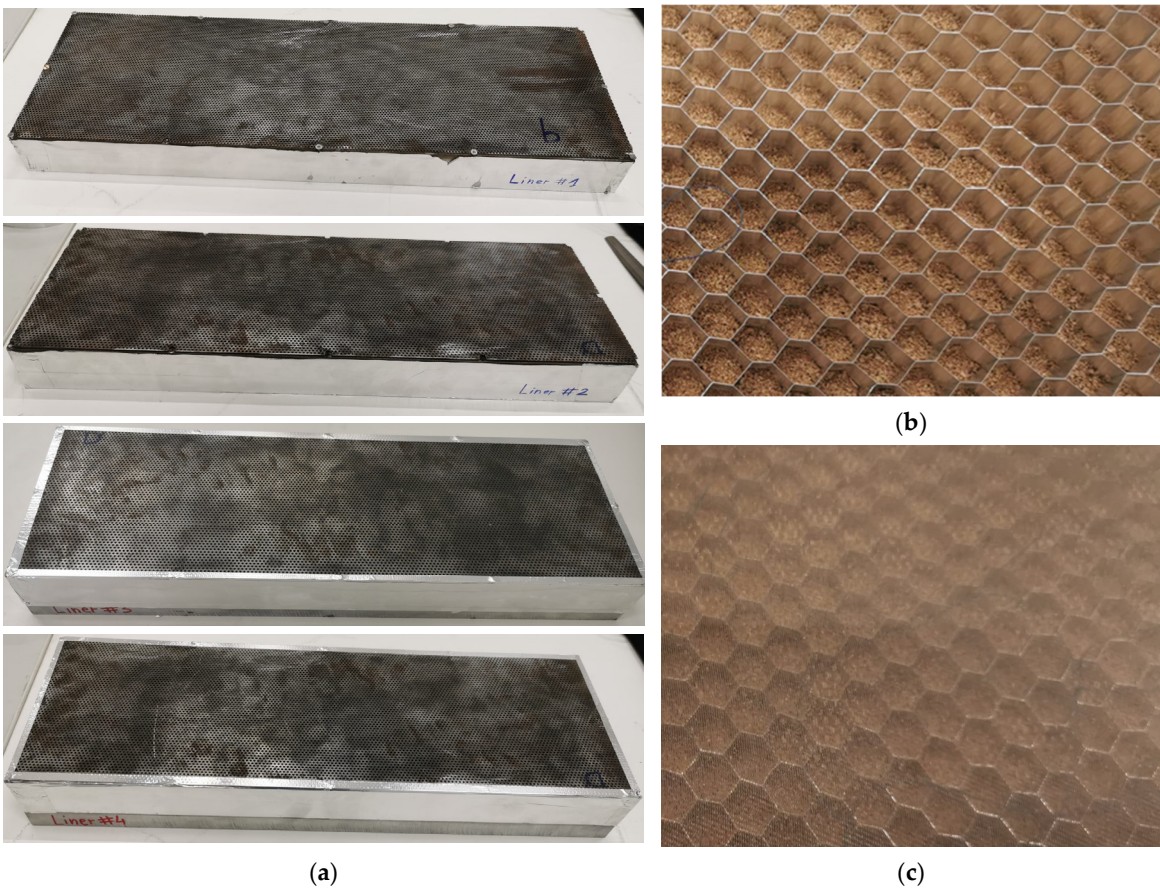

(a)   (b)   (c)

**Figure 7.** Liner overview: (**a**) tested configurations; (**b**) 66% cork filled honeycomb without mesh; (**c**) 66% cork filled honeycomb without mesh.

Similar to Figures 8 and 9, the insertion loss recorded on the test bench is maintained over the entire frequency range (500–6000 Hz) for several flow speeds. The efficiency of powder filling is evident at low speeds and it subsides at high Mach numbers. The highest efficiency, for these configurations, is found in the 1000–5000 Hz interval, which corresponds to the interval in which the BPF and its first harmonics are located. The influence of the mesh placed between the honeycomb and the cover is not significant, from Figure 9b, and it cannot be negatively qualified from an acoustic point of view (the same overlap of the graphs was obtained for the other test configurations). Even the holes obstruction on the contour of the plates did not have an undesirable effect, but the ratio between the obstructed area and the free area is very small.

The graphs in Figures 8 and 9 were determined by comparing the spectrum (transmission loss) obtained with the new configurations in the HW (hard wall) variant where, instead of perforations, a smooth aluminum plate of the same thickness was used, where the only attenuation effect was given by the interaction between the propagating wave and the wall reflections. An example of gross TL is highlighted in Figure 10.

The IL data (Figure 8) highlight attenuations of 2–3 dB of the liner samples filled with powders, on certain frequency values of up to 5 dB. The introduction of a mesh at the interface between the honeycomb and the perforated cover leads to a slight increase in acoustic attenuation as a result of the increased friction of the air particles inside the sheet perforation. To better quantify the noise reduction, the graphs should have been made with a lower resolution (a moving average function could have been used). Also, global levels should be calculated over different frequency ranges to avoid too many interpretations. Table 1 shows a logarithmic averaging of the values in Figure 8a–d.

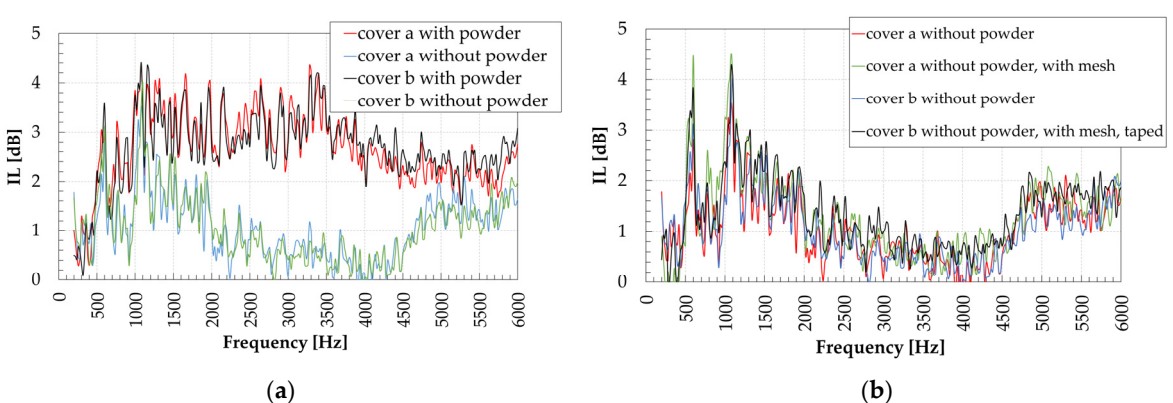

**Figure 8.** Insertion loss (covers with powder and mesh): (**a**) Liner 1; (**b**) Liner 2; (**c**) Liner 3; (**d**) Liner 4.

**Figure 9.** *Cont*.

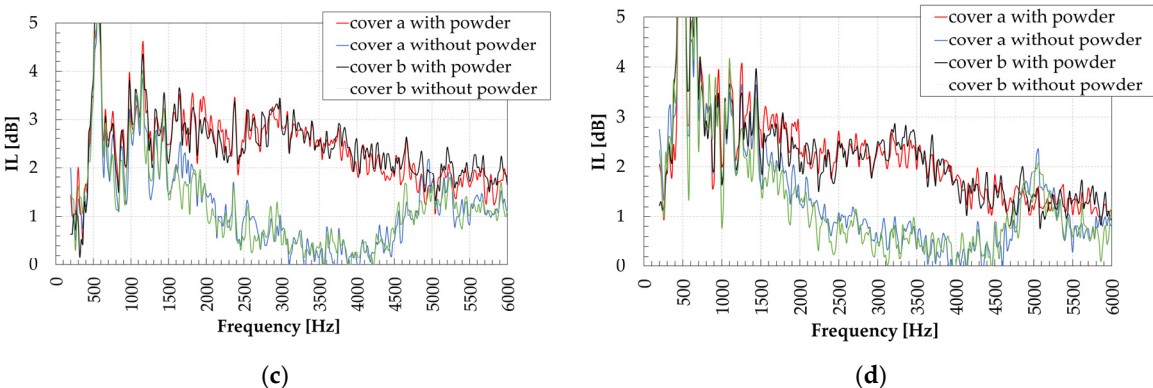

(**c**)                                          (**d**)

**Figure 9.** Cork powder influence: (**a**) Liner 4, Mach 0; (**b**) Liner 4, Mach 0—empty liner with or without mesh or tape around the corners (7–10 mm); (**c**) Liner 4, Mach 0.2; (**d**) Liner 4, Mach 0.4.

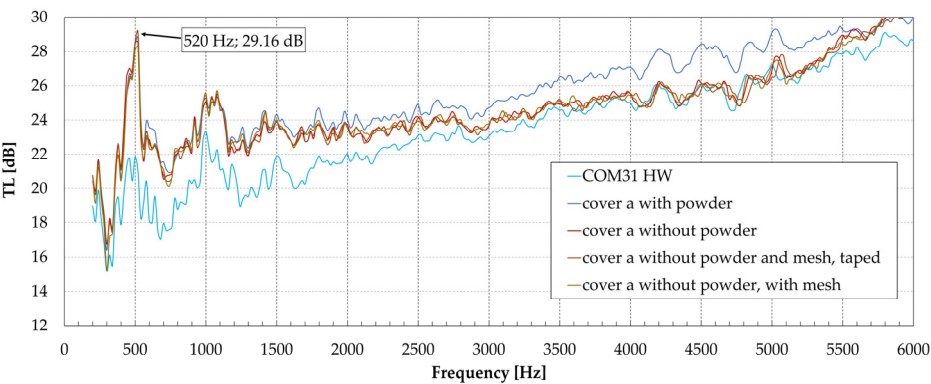

**Figure 10.** Transmission loss for liner 3, Mach 0.4.

**Table 1.** Logarithmic averaging of insertion loss.

|  |  | Liner 1 | | | Liner 2 | | | Liner 3 | | | Liner 4 | | |
|---|---|---|---|---|---|---|---|---|---|---|---|---|---|
|  |  | M = 0 | M = 0.2 | M = 0.4 | M = 0 | M = 0.2 | M = 0.4 | M = 0 | M = 0.2 | M = 0.4 | M = 0 | M = 0.2 | M = 0.4 |
| Cover A | global (0–6 kHz) | 1.75 | 1.68 | 1.79 | 2.05 | 1.90 | 2.09 | 2.38 | 2.29 | 2.30 | 2.72 | 2.47 | 2.32 |
| | 200–2000 Hz | 1.76 | 2.10 | 2.71 | 2.25 | 2.44 | 2.97 | 2.40 | 2.71 | 3.25 | 2.77 | 2.88 | 3.30 |
| | 2000–6000 Hz | 1.75 | 1.48 | 1.31 | 1.95 | 1.64 | 1.62 | 2.37 | 2.08 | 1.79 | 2.70 | 2.27 | 1.80 |
| Cover B | global (0–6 kHz) | 1.81 | 1.73 | 1.91 | 2.45 | 2.14 | 1.94 | 2.52 | 2.36 | 2.20 | 2.75 | 2.51 | 2.46 |
| | 200–2000 Hz | 1.82 | 2.20 | 2.76 | 2.32 | 2.61 | 2.77 | 2.34 | 2.72 | 3.23 | 2.65 | 2.86 | 3.60 |
| | 2000–6000 Hz | 1.82 | 1.50 | 1.47 | 2.50 | 1.92 | 1.51 | 2.60 | 2.19 | 1.64 | 2.80 | 2.35 | 1.81 |

The acoustic differences produced by cover A and cover B are small, with the main parameter affecting the attenuation performance being the Mach number and its increase leading to a decrease in attenuation at high frequencies, which is justified by the fact that the static pressure of the air flow decreases the reactance of the liner at high frequencies. On the other hand, it can be seen that the influence of porosity is small for low speeds (M = 0 and M = 0.2) and this influence decreases for higher speeds (M = 0.4).

The insertion of powder into the cavities leads to a slight increase in the attenuation in the 1500–4500 Hz frequency domain, which can be produced both by the friction phenomenon of the particles but also by the fact that the powder behaves as a porous acoustic material that leads to the lamination of the air molecules among the powder particles—the classic case of porous material (Figure 9). However, seeing the last measurements made after the ARTEM project when extremely fine powders were tested in Kundt tubes which led to a very high absorption for low frequencies, we expect that another phenomenon

occurs: friction of air against the rough and irregular external surface of powder particles. It seems that under the influence of sound, particles occupy the whole volume of the honeycomb generating a mixture which has an increased capacity to absorb sound. Looking at the graphs from Figure 10 one can observe an increase in transmission loss under 1500 Hz when the powder is used relatively to the HW (hard wall) case. As the cork powder is a hydrophobic material and its holding inside the liner is achieved by means of an acoustically permeable mesh, the solution can be very useful if it is applied to gas turbine engine intake ducts, a subject being intensively researched at present.

A first model would be the decomposition method, whereby sound pressure can be decomposed into incident and reflected waves by the two-load method or the two-source method (Figure 11).

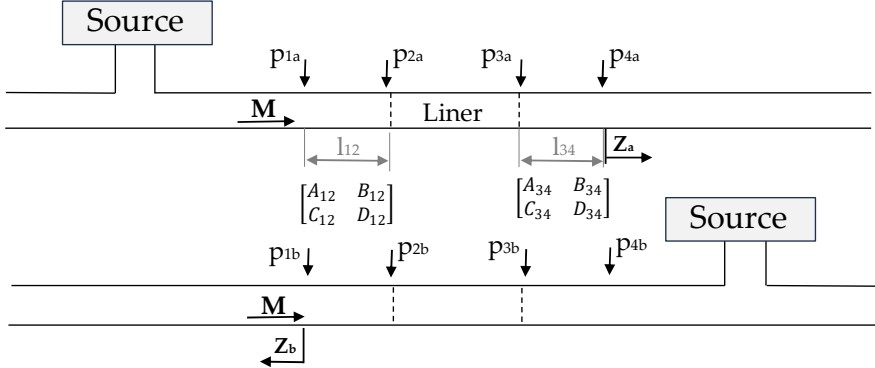

**Figure 11.** Two-source method.

The transfer matrix of the region where the liner is placed is given by Equation (1)

$$\begin{bmatrix} p_1 \\ v_1 \end{bmatrix} = \begin{bmatrix} A & B \\ C & D \end{bmatrix} \begin{bmatrix} p_2 \\ v_2 \end{bmatrix} \tag{1}$$

where the source is on the left side of the wave propagating to the right.

For the case of direct flow (left to right), one can write the set of Equations (2)–(5)

$$\begin{bmatrix} p_{1a} \\ v_{1a} \end{bmatrix} = \begin{bmatrix} A_{12} & B_{12} \\ C_{12} & D_{12} \end{bmatrix} \begin{bmatrix} A_{23} & B_{23} \\ C_{23} & D_{23} \end{bmatrix} \begin{bmatrix} A_{34} & B_{34} \\ C_{34} & D_{34} \end{bmatrix} \begin{bmatrix} p_{4a} \\ p_{4a}/Z_a \end{bmatrix} \tag{2}$$

$$\frac{p_{3a}}{p_{4a}} = A_{34} + \frac{B_{34}}{Z_a} = H_{34a} \tag{3}$$

where $H_{ij} = \frac{p_i}{p_j}$ is the transfer function

$$\frac{p_{2a}}{p_{4a}} = A_{23}A_{34} + \frac{A_{23}B_{34} + B_{23}B_{34}}{Z_a} = H_{24a} \tag{4}$$

$$\frac{p_{1a}}{p_{4a}} = A_{12}(A_{23}A_{34} + B_{23}C_{34}) + B_{12}(C_{23}A_{34} + D_{23}C_{34}) \\ + \frac{A_{12}(A_{23}B_{34} + B_{23}D_{34}) + B_{12}(C_{23}B_{34} + D_{23}D_{34})}{Z_a} = H_{14a} \tag{5}$$

The reverse flow reduces to Equation (6).

$$\begin{bmatrix} p_2 \\ -v_2 \end{bmatrix} = \frac{1}{\Delta} \begin{bmatrix} D_{23} & B_{23} \\ C_{23} & A_{23} \end{bmatrix} \begin{bmatrix} p_1 \\ -v_1 \end{bmatrix} \tag{6}$$

where $\Delta = A_{23}D_{23} - B_{23}C_{23}$ is the determinant of the matrix.

By inserting in the transfer matrices for the three domains and substituting some terms in the resulting equations, the propagation in the microphone area can be written:

$$\begin{bmatrix} A_{12} & B_{12} \\ C_{12} & D_{12} \end{bmatrix} = e^{-M(jk_c + \propto_c)l_{12}} \begin{bmatrix} cosh\beta_{12} & Ysinh\beta_{12} \\ sinh\beta_{12}/Z & cosh\beta_{12} \end{bmatrix} \tag{7}$$

$$\begin{bmatrix} A_{34} & B_{34} \\ C_{34} & D_{34} \end{bmatrix} = e^{-M(jk_c + \propto_c)l_{1342}} \begin{bmatrix} cosh\beta_{34} & Ysinh\beta_{34} \\ sinh\beta_{34}/Z & cosh\beta_{34} \end{bmatrix} \tag{8}$$

where $\Delta_{12} = e^{-2M(jk_c+\propto_c)l_{12}}$, $\Delta_{34} = e^{-2M(jk_c+\propto_c)l_{34}}$, $\alpha_c = \frac{\alpha}{(1-M^2)}$, $k_c = \frac{k}{(1-M^2)}$, and $l_{12}$ and $l_{34}$ is the distance between microphones. In case of neglecting the flow, one can write $\Delta_{12} = 1$ and $\Delta_{34} = 1$ and TL is written according to Equation (9).

$$TL = 20lg\left[\frac{1}{2}\left|A_{23} + \frac{B_{23}}{Z} + ZC_{23} + D_{23}\right|\right] \tag{9}$$

For the liner impedance calculation, one can also use the theories of Kooi and Sarin [22], Guess [23], and Maa [24], adjusted to flows with relatively large Mach numbers.

The good initial results determined the authors to continue researching this physical phenomenon and the technology which is very promising for applications in aerospace because it offers a new possibility to absorb broadband noise in the low frequency range. Furthermore, during the research it was observed that a reduction in the granule's dimensions can lead to a higher noise absorption (e.g., absorption coefficient in the case of Ø1.3 mm polystyrene balls is higher than in the case of Ø2.3 mm polystyrene balls [7]). Thus, after the project, the manufacturing technology of these powders was changed in order to go with smaller cork powder granules and the impact grinding technology (a sandpaper grinding technology) was chosen. Five types of sandpaper were used: P40, P60, P80, P120 and P180. Some images and average granule dimensions of these powders are given in Figures 12 and 13.

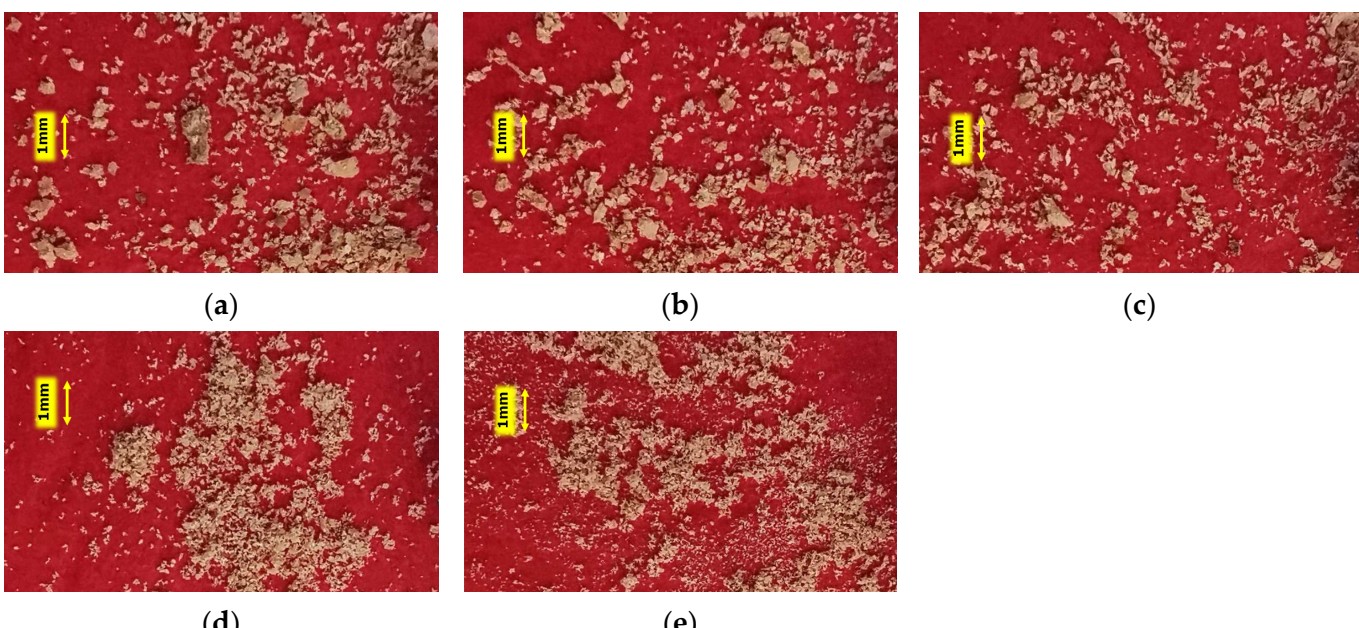

**Figure 12.** Fine powders manufactured with sandpaper grinding technologies (**a**) P40; (**b**) P60; (**c**) P80; (**d**) P120; (**e**) P180.

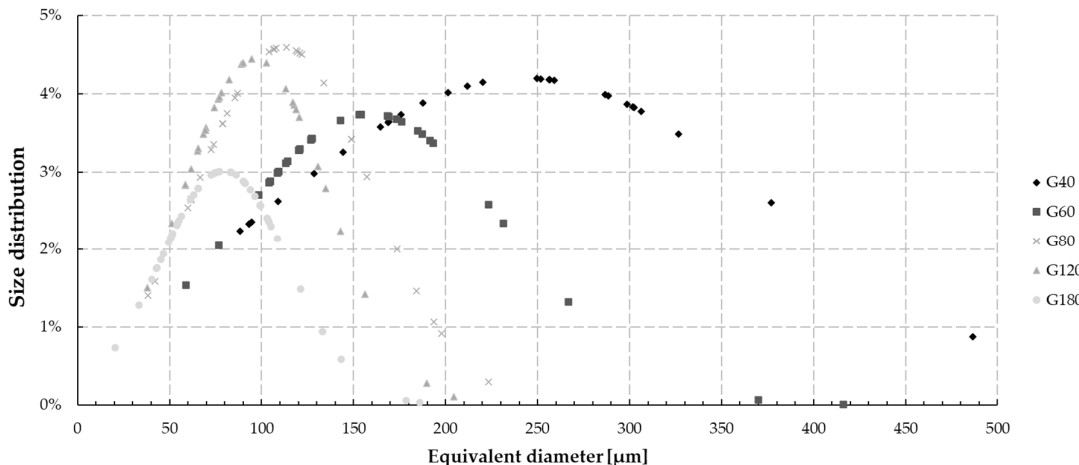

**Figure 13.** Cork powder size distribution.

All these different sized powders were further tested in the Kundt tube using Φ28 mm honeycomb samples in different combinations with various types of cover (perforated aluminum plate: porosity 28.5%, $\varnothing_{hole}$ 1.8 mm, thickness # 0.7 mm/perforated steel plate, porosity 33%, thickness # 1.5 mm) and three different meshes in order to prevent the cork coming out. As powder sizes tend to decrease to enhance the friction effect, a solution is needed to prevent the powder from escaping from the cavities. For G40-G180 granules, three acoustically permeable meshes with different weaves were placed between the cover and the cavity. Because the original tests (Kundt tube) resulted in good absorptions for both 66% and 100%, testing was continued with these two configurations (at least for comparison, even though the greatest effect should be with partial filling, as some energy is dissipated by particle friction). The results (Figure 14) show an increase in the absorption of low frequency noise in comparison with the noise absorption of cork powder with larger grain diameters (as the one used in Le Mans and NLR, where the average size of the particles was around 800 μm for cork powder, which was manufactured using an impact grinding technology).

The tested powders show a broadband absorption in the given test conditions (Kundt tube, normal incident wave). There is a slight difference between the use of aluminum vs. steel covers (the most noticeable being in Figure 14e), but the factor that increases α is the cavity filling. It seems that the effects of friction between the particles together with the particle size increase the performance of the liner. The meshes referred to in Figure 14 were analyzed under a microscope (Figure 15) and their structure, as well as the dimensions, were quite different. The fact that they are almost as acoustically permeable presents an advantage in terms of combining them with a larger range of powder granules. The studies regarding the placement of a fabric glued to the perforated panel led to the increase of the friction phenomenon in the neck of the resonator, which implicitly leads to the dissipation of the acoustic energy, obtaining a higher acoustic absorption. As a rule, the thicker the fabric, the greater the friction, this phenomenon being influenced by the filling degree of the cavities. The fuller the cavity, the lower the flow of air going in and out (the counter pressure is higher), the reactance decreases and the absorption is lower, as meshes with larger openings are needed.

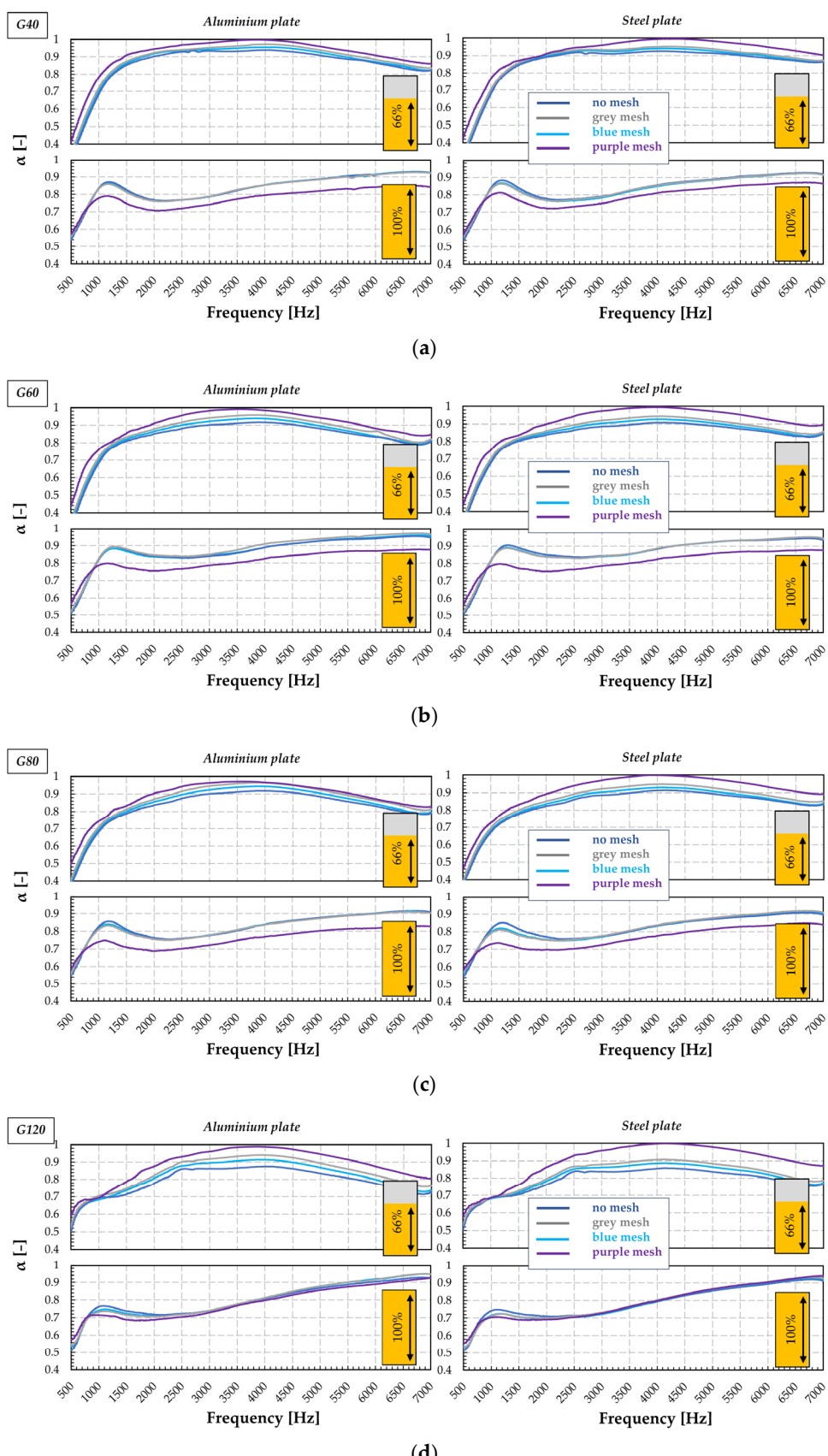

**Figure 14.** *Cont.*

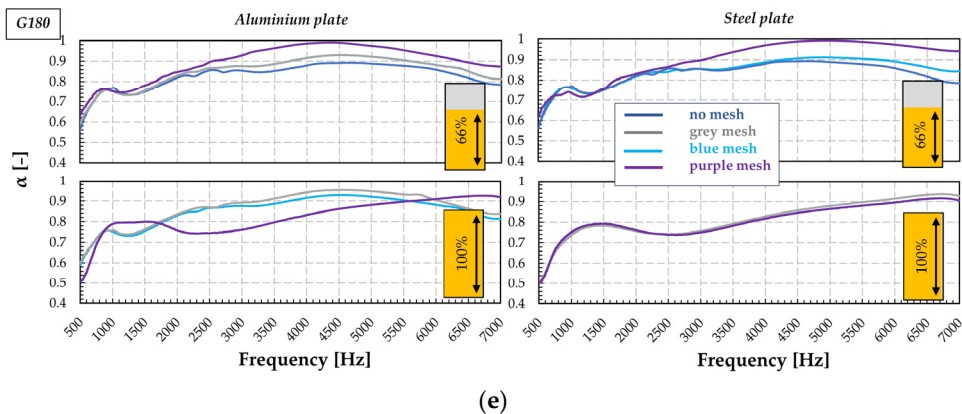

(**e**)

**Figure 14.** Noise absorption (alpha coefficient) for various cork granulations: (**a**) G40; (**b**) G60; (**c**) G80; (**d**) G120; (**e**) G180.

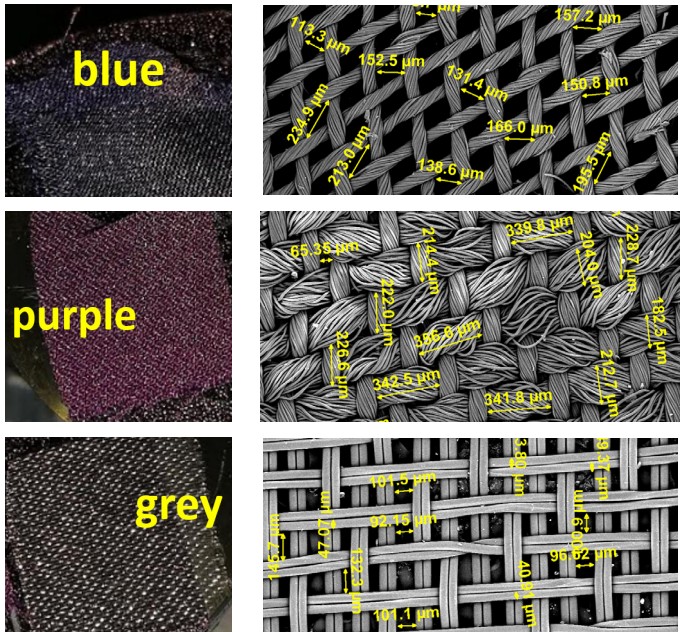

**Figure 15.** Acoustic permeable meshes.

## 3. Conclusions

The preliminary tests carried out on different powders showed that the introduction of such materials into the resonant cavities of acoustic liners improves their absorption. Although many materials seemed promising, the conditions under which they work must also be taken into account. The cork powder turned out to be the best, both acoustically and from the point of view of applicability and availability (being accessible and waterproof). It has been proven that the cavity filling percentage plays an important role, with the optimal value analyzed within the project being around 2/3. In this version, the new air-powder medium behaves like another material, most likely due to the collisions and "frictions" between the particles. It was observed that as the size of the particles (statistically) decreases, the acoustic destructive phenomenon increases.

Future research will be directed towards the analysis of particle movement inside a resonant cavity (rather than obtaining a global flow resistivity parameter) using high fps video recording means. An analytical approach to the phenomenon will also be attempted. Future research will also investigate the effect of the increase of the acoustic absorption towards low frequencies (around 500 Hz) that occurs with the decrease in the average size of the powders.

**Author Contributions:** Conceptualization, C.S.; methodology, C.S., A.-G.T., A.-T.T. and M.D.; validation, C.S., A.-G.T., A.-T.T. and M.D.; formal analysis, C.S., A.-G.T. and M.D.; investigation, C.S., A.-G.T., A.-T.T. and M.D.; data curation, A.-G.T. and M.D.; writing—original draft preparation, C.S. and A.-T.T.; writing—review and editing, C.S., A.-G.T., A.-T.T. and M.D.; visualization, A.-G.T.; supervision, C.S.; project administration, C.S. All authors have read and agreed to the published version of the manuscript.

**Funding:** This research was funded by Horizon2020, grant number 769350.

**Data Availability Statement:** Data available upon written request to the corresponding author.

**Acknowledgments:** The data presented and analyzed in this report were obtained with the help of COMOTI's Research and Experiments Center in the field of Acoustic and Vibrations staff and facility.

**Conflicts of Interest:** The authors declare no conflict of interest.

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
