# Peer review of "Advances in Experimental Research on the Influence of Friction Powders on Acoustic Liners (Helmholtz Resonators)"

_aerospace, doi:10.3390/aerospace10121000_

Round 1
Reviewer 1 Report
Comments and Suggestions for Authors
Title: Advances in Experimental Research on the Influence of Friction Powders on Acoustic Liners (Helmholtz Resonators)
In this article to present the technological advancement of using friction powders to increase the absorption of acoustic liners used in the reduction of tonal noise, a total of 17 various materials in Kundt impedance tubes are tested. The noise absorption method is creative, and the topic is interesting, some minor problems need to be revised.
1. It is better to replace the Figure 5 with a clearer version.
2. Why did you choose Insertion Loss instead of directly selecting noise reduction levels as the evaluation indicator, dose It have any special meaning? The meaning of Insertion Loss needs a further elaboration in order to make it clearer to the readers.
3. In my opinion, “The highest efficiency, for these configurations, is found in the 1000-5000Hz interval” in line 159 is not very consistent with the fact shown in Figure 7. Some peaks appear near 500Hz, please give a more detailed statement of the underlying reason for this situation.
4. In Figure 8 except for Figure8(b), there is a large deviation between cover a and cover b in the frequency range from 1500Hz to 4500Hz. What is the cause of this deviation and how does it affect the conclusion reached in this paper?
5. It is suggested to add some mathematical analysis of experimental data.
Comments on the Quality of English Languagenone
Author Response
Hello,
Thank you very much for taking the time to review this manuscript. I have tried to respond to all the comments received on the article. I will list what has been changed/added at each point:
1-Figure 5 has been replaced (Line 170)
2-Some clarifications in lines 153-168
3-Some comment in lines 204-207 + Table1
4-Some clarifications in lines 217-227
5-An overview of analytical models is presented in lines 234-250; Since the experimental phase, a mathematical model is being worked on to capture most of the CNRS and NLR results. The problem is, the models listed in line 247 give (at best) slightly different results from certain configurations. The consistency of the models at high speeds is still questionable for the powder solution.
We hope that the changes made will clarify a little more the information presented, the aspects you have brought up have been debated with interest and we intend to be more concise in the following papers (the influence of particle friction being an aspect scarcely addressed in the literature)

Reviewer 2 Report
Comments and Suggestions for Authors
The authors studied acoustic liners with the cavities filled with various types and quantities of powders and demonstrated their acoustic performance. This technology is interesting and promising for aero-engine noise reduction. Here are some comments
1- What do you mean by COMOTI in the abstract, EPFL and ISVR in the introduction in page 1 ?
2-How many microphones do you use and what are the sensibilities of these microphones ? Since multiple microphones are shown in figure 3, how do you calibrate all of these microphones? How do you measure the transmission loss ? Do you use two-sources or tow loads methods ?
3-For the nacelle liners where the porosity of the micro perforated panel (MPP) is lower than 15%, can the powder in the cavities increase the sound attenuation? The porosity that the authors consider is higher than 20%. If you consider a liner with a MPP porosity of 5% for example, do you think that its acoustic attenuation can be increased by filling the cavity by powders? What is the porosity limits to use the powders in the cavity in order to improve the sound attenuation performance?
4-Are the powders fire resistant? For aerospace application, apart from the acoustic performance of this technology, there will be a need of fire test.
5-In the introduction, the authors should also mention the liner technology with the cavity filled with bulk materials and give some references
Author Response
Hello,
Thank you very much for taking the time to review this manuscript. I have tried to respond to all the comments received on the article. I will list what has been changed/added at each point:
1- COMOTI is the abbreviation in Romanian for Romanian Research and Development Institute for Gas Turbines, it is also mentioned in the authors' affiliation. EPFL (École Polytechnique Fédérale de Lausanne, Switzerland) and ISVR (Institute of Sound and Vibration Research, Southampton) have been added to the text (Lines 37-38; Line 42)
2-Two references have been made to papers detailing the test stand used in the project (line 134). Related to the measurement method, the analytical calculation procedure was also briefly presented (lines 234-250).
3-Some words on lines 172-176; From what experiments have been done in the Kundt tube, the porosity does not affect so much the value of the absorption coefficient, the lowest value experimented being 8.37%. The case with grazing flow at such a configuration has not been addressed in the project but is foreseen in future research.
4-As the acoustic liners addressed in the project were intended for application on the intake, their fire resistance was not an issue. The environment in which they perform is at worst a wet one, therefore the solution that has been determined to be optimal is the use of cork powder which besides good acoustic properties is suitable for moist environments (unlike balsa, ash and talcum powder). In the case of adopting such a solution on the hot area of the engine, the research we are currently carrying out considers the use of powders of fire-resistant fibres (rockwool, fiberglass), empty small glass spheres, aerogel powder differently from those experimented in the project and require other cavity filling height-plate porosity combinations.
5-Some words on lines 58-63
We hope that the changes made will clarify a little more the information presented, the aspects you have brought up have been debated with interest and we intend to be more concise in the following papers (the influence of particle friction being an aspect scarcely addressed in the literature)

Reviewer 3 Report
Comments and Suggestions for Authors
The manuscript reports a experimental work on the effect of different friction powders on the acoustic liners using Helmholtz resonators. It also investigates the performance of the liners using various covers of the resonators.
The work shows impressive performance increase in the sound absorption using friction powders in the resonators. However, I cannot recommend its publication due to some unclear descriptions in the experimental setup and results as follows.
1. In Line 110 (also 130), the powders were said to fill up 33%, 66% and 100% of the resonator heights. However, only the results with 66% and 100% are shown. In this case, the 33% set up can be omitted.
2. The Mach number mentioned in the setup (Line 131) are 0 and 0.13. However, the authors presented results with Mach number 0.2 and 0.4 in Figure 8.
3. What are the configurations of Liner # 1 - 4? I can only find their difference in liner height (Fig. 5). However, what are other differences? The authors may consider using a table to list out all the configurations of the liners and test conditions
4. The authors may also consider detailing the configurations before the discussion of the results. For instance, readers have difficulties in understanding the blue, purple and grey meshes when they present their results.
Comments on the Quality of English Language
I can understand the English without any issues.
Author Response
Hello,
thank you for your effort and interest in reviewing this article. I have tried to respond as best I can to the comments and the following are the punctual additions/modifications/justifications made (in text with purple):
1- A series of powders were tried in the Kundt tube at fillings of 33, 66 and 100% of the liner height, as mentioned in line 115. Of these variants, only 6 were selected (line 132) and tested at CNRS (lines 133-136) under grazing flow conditions. In line 130 a reference is also made to an article [7] with more detailed results from that test campaign. The CNRS tests resulted in a material that performs very well when the liner is placed in flow (cork powder) and was subsequently tested at NLR at a slightly larger scale (with a percentage fill, this time, of only 66% - this was determined as the optimum - line 137). In line 253 the rationale for continuing the tests (after NLR) is presented and because in the original tests (Kundt tube) good absorption was obtained for both 66% and 100%, testing was continued with these two configurations (at least for comparison, even though the greatest effect should be with partial filling, as some of the energy is dissipated by particle friction). --- last part was added in lines 275-278
2-In the introductory part (lines 114-141), the steps taken in the project are presented in chronological order. The second campaign (after Kundt tube tests, i.e. without flow) was carried out at CNRS-Le Mans at speeds corresponding to Mach 0 and 0.13. Subsequently, the good results at CNRS (and presented in [7]) led to NLR tests at other speeds (Mach 0.2 and 0.4 added in the text at line 140). As the CNRS results have already been presented, the article is based on the presentation of the NLR tests, while retaining a reference to the original work.
3-As you well noticed, the only difference regarding liners 1-4 is the height (added as an observation to line145). The final configurations are obtained by combining the different heights with the 2 covers, as mentioned on line 178. In order not to allow the powders to escape from the cavity (although in these tests there is not necessarily such a risk, as they are not placed with the cover down) some configurations were obtained with a certain height of liner, cover and mesh (also mentioned in lines 180-182). The results in figure 8, as can be seen from the figure description, show the results at 3 flow Mach numbers for the 4 liners in combination with the 2 covers (3 columns x 4 lines x 2curves). To highlight that the mesh does not influence the liner performance, several plots have been plotted in Figure 9.
4-Some clarifications on meshes have been added to lines 272-275
I took the liberty of leaving the other comments for reviewers 1 and 2 (highlighted in yellow and blue) in which I hope I have clarified some of the issues you mentioned.I hope that the additions made will clarify the work a little more. The aspects that you mentioned have generated constructive discussions and we will take them into account in the works that will follow, as this solution of using powders to increase the noise reduction capabilities is scarcely addressed in the specialized literature.

Round 2
Reviewer 2 Report
Comments and Suggestions for Authors
The authors have addressed all the comments